# Al[^18^F]F-NOTA-Octreotide Is Comparable to [^68^Ga]Ga-DOTA-TATE for PET/CT Imaging of Neuroendocrine Tumours in the Latin-American Population

**DOI:** 10.3390/cancers15020439

**Published:** 2023-01-10

**Authors:** Arlette Haeger, Cristian Soza-Ried, Vasko Kramer, Ana Hurtado de Mendoza, Elisabeth Eppard, Noémie Emmanuel, Johanna Wettlin, Horacio Amaral, René Fernández

**Affiliations:** 1Nuclear Medicine and PET/CT Center PositronMed, Providencia, Santiago 7501068, Chile; 2Positronpharma SA, Providencia, Santiago 7500921, Chile; 3Department of Nuclear Medicine, University Hospital Magdeburg, 39120 Magdeburg, Germany; 4Ion Beam Applications, 1348 Louvain-la-Neuve, Belgium

**Keywords:** neuroendocrine tumours, SSTR, Al[^18^F]F-NOTA-Octreotide, PET imaging

## Abstract

**Simple Summary:**

In the present work we investigated the clinical utility of Al[^18^F]F-NOTA-Octreotide (Al[^18^F]F-OC) in comparison to [^68^Ga]Ga-DOTA-TATE in patients diagnosed with neuroendocrine tumours. Our aim was to verify the recently published, promising results for Al[^18^F]F-NOTA-Octreotide in the Latin-American population. Al[^18^F]F-NOTA-Octreotide provided excellent image quality, detected NET lesions with high sensitivity and represents a highly promising, clinical alternative to [^68^Ga]Ga-DOTA-TATE.

**Abstract:**

PET imaging of neuroendocrine tumours (NET) is well established for staging and therapy follow-up. The short half-life, increasing costs, and regulatory issues significantly limit the availability of approved imaging agents, such as [^68^Ga]Ga-DOTA-TATE. Al[^18^F]F-NOTA-Octreotide provides a similar biodistribution and tumour uptake, can be produced on a large scale and may improve access to precision imaging. Here we prospectively compared the clinical utility of [^68^Ga]Ga-DOTA-TATE and Al[^18^F]F-NOTA-Octreotide in the Latin-American population. Our results showed that in patients with stage IV NETs [^68^Ga]Ga-DOTA-TATE presents higher physiological uptake than Al[^18^F]F-NOTA-Octreotide in the liver, hypophysis, salivary glands, adrenal glands (all *p* < 0.001), pancreatic uncinated process, kidneys, and small intestine (all *p* < 0.05). Nevertheless, despite the lower background uptake of Al[^18^F]F-NOTA-Octreotide, comparative analysis of tumour-to-liver (TLR) and tumour-to-spleen (TSR) showed no statistically significant difference for lesions in the liver, bone, lymph nodes, and other tissues. Only three discordant lesions in highly-metastases livers were detected by [^68^Ga]Ga-DOTA-TATE but not by Al[^18^F]F-NOTA-Octreotide and only one discordant lesion was detected by Al[^18^F]F-NOTA-Octreotide but not by [^68^Ga]Ga-DOTA-TATE. Non-inferiority analysis showed that Al[^18^F]F-NOTA-Octreotide is comparable to [^68^Ga]Ga-DOTA-TATE. Hence, our results demonstrate that Al[^18^F]F-NOTA-Octreotide provided excellent image quality, visualized NET lesions with high sensitivity and represents a highly promising, clinical alternative to [^68^Ga]Ga-DOTA-TATE.

## 1. Introduction

Neuroendocrine tumours (NETs) are highly heterogeneous neoplasms that arise from neuroendocrine cells and affect the diffuse neuroendocrine system, intestinal tract and bronchia [1]. The slow progression of these tumours and their unspecific symptoms lead to a high prevalence and late diagnoses [2,3,4]. Unfortunately, late diagnoses favour the development of distant metastasis resulting in high mortality [5]. To improve the clinical outcome for NET patients, early detection of these tumours before dissemination is of utmost importance. NETs are characterised by the high expression of somatostatin receptors (SSTR) [6], chromogranin A and synaptophysin [7]. SSTR is a G protein-coupled receptor that binds the somatostatin neuropeptide, which is paracrine secreted by gastrointestinal and brain cells. Although there are five SSTR subtypes (SSTR 1–5), 80% of NETs overexpress SSTR2 [8]. Diagnostic imaging, staging, and follow-up after treatment can be performed by single-photon emission computed tomography (SPECT/CT) or positron emission tomography/computed tomography (PET/CT) with radiolabelled somatostatin analogues [9]. To this end, radiotracers such as [^68^Ga]Ga-DOTA-TATE, [^68^Ga]Ga-DOTA-TOC, and [^68^Ga]Ga-DOTA-NOC are the standard radiopharmaceuticals for NET detection [10,11,12]. PET imaging of SSTR is further required for pretherapeutic evaluation of patients who are candidates for SSTR-targeted therapy with [^177^Lu]Lu-DOTA-TATE [13].

However, the high cost of ^68^Ge/^68^Ga generators [9], the short half-life of Galium-68 (68 min), regulatory and quality assurance aspects significantly limit the availability of these tracers, especially in Latin-America and developing countries. Fluorine-18 has a longer half-life (109.8 min) and a better spatial resolution as compared to Gallium-68 due to the lower positron-energy [9,14,15]. Several ^18^F-labelled SSTR-radioligands have been developed in recent years which can be produced in large scale and satellite-distribution over longer distances is feasible [16]. Al[^18^F]F-NOTA-Octreotide is one of the most promising candidates [14,15,17,18], providing a high affinity for SSTR2 [17,19], a favourable biodistribution, high tumour uptake and has proven to be safe in clinical applications [15].

Several studies have compared Al[^18^F]F-NOTA-Octreotide and [^68^Ga]Ga-DOTA-TATE PET imaging in NET patients showing outstanding results [14,20,21,22]. A first comparison was published in a case report by Pauwels et al. in 2019 and, despite their different chemical structures (Appendix A), both ligands showed a very similar biodistribution in healthy organs and NET lesions [20]. These initial results were confirmed in a systematic evaluation of six NET patients by the same group[21] and independently in a study conducted by Hou et al. including 20 NET patients [14]. Both studies proofed the non-inferiority of Al[^18^F]F-NOTA-Octreotide and showed even higher tumour-to-liver ratios (TLR) as compared to [^68^Ga]Ga-DOTA-TATE [14,21]. Finally, a recent, prospective study in 75 patients found that Al[^18^F]F-NOTA-Octreotide outperformed [^68^Ga]Ga-DOTA-TATE showing significantly higher detection rates and tumour-to-background ratios when PET images were acquired two hours post injection [22]. Given the limited access to [^68^Ga]Ga-DOTA-TATE in vast countries, such as Chile, in this prospective study we compared the clinical utility of Al[^18^F]F-NOTA-Octreotide PET/CT imaging with [^68^Ga]Ga-DOTA-TATE PET/CT and clinical parameters of NET patients in the Latin-American population.

## 2. Materials and Methods

### 2.1. Radiochemistry

Al[^18^F]F-NOTA-Octreotide was produced in accordance with local GMP-regulations using a modified procedure similar to a process previously described [23] and as detailed in the Appendix A. Briefly, 35 ± 19 GBq (range 8.1–54.6 GBq) Al[^18^F]F-NOTA-Octreotide were obtained from starting activities of 100 ± 51 GBq (range: 40–170 GBq) as a sterile solution after 31 min in 33.7 ± 8.9% (N = 8) radiochemical yield (n.d.c.), >95% radiochemical purity, and specific activities of 114 ± 61 GBq/μmol. For further details regarding specifications and results, see Appendix A.

### 2.2. PET/CT Imaging

A total of 20 patients (age: 57.3 ± 11.1 y) with biopsy-proven, stage IV NET and complying with all inclusion criteria (see Appendix A) were enrolled in this prospective study (Table 1). Al[^18^F]F-NOTA-Octreotide and [^68^Ga]Ga-DOTA-TATE were injected intravenously at a dose of 222–296 MBq and 148–185 MBq, respectively, and PET/CT images were acquired head-to-mid-thigh at 60 ± 10 min post injection (Biograph Vision, Siemens, Erlangen, Germany). In some cases, we had protocol deviations with lower injected activities (Table 1). The time interval between both PET/CT scans was 12.7 ± 8.0 days (range: 2–30 days), without any treatment during the interval. A low dose CT and contrast-enhanced CT scan was performed for anatomical localisation and attenuation correction for Al[^18^F]F-NOTA-Octreotide and [^68^Ga]Ga-DOTA-TATE PET/CT, respectively.

### 2.3. Image Analysis

Volumes of interest (VOIs) were drawn around tumour lesions, visually distinguished as regions of increased radiotracer uptake relative to background uptake and expected physiological radiotracer uptake. To perform semi-quantitative analysis, mean, peak and maximum standard uptake values (SUV_bw_) were calculated using Siemens SyngoVia software. Two nuclear medicine experts who were not blinded to clinical data, independently analysed the PET images. The biodistribution profiles in normal organs were compared for both tracers by analysing SUV_mean_ and SUV_max_ values. Likewise, SUV_max_ and SUV_peak_ values were used to compare the uptake of Al[^18^F]F-NOTA-Octreotide and [^68^Ga]Ga-DOTA-TATE in NET lesions. Tumour-to-liver (TLR) and tumour-to-spleen ratios (TSR) were calculated by dividing the SUV_max_ of different tumour lesions by the SUV_mean_ of the liver and spleen, respectively.

### 2.4. Statistical Analysis

Continuous variables were evaluated for normal distribution with histograms and Q-Q plots. Nonparametric quantitative data were compared using a two-sided Wilcoxon signed-rank test to analyse and compare SUV, TLR, and TSR values between scans with *p*-values < 0.05 considered as statistically significant. To test the non-inferiority of Al[^18^F]F-NOTA-Octreotide compared to [^68^Ga]Ga-DOTA-TATE, malignant lesions detected in each patient were registered and counted. In the case of an excessive number of lesions (≥50) an arbitrary number of 50 lesions was used. Since cancerous lesions within a subject are likely to be more correlated than cancerous lesions between subjects, a linear mixed-effects model of non-inferiority for repeated measures was employed [24]. The study had a clinically significant non-inferiority margin of 5% to show a non-inferiority of Al[^18^F]F-NOTA-Octreotide compared to [^68^Ga]Ga-DOTA-TATE in tumoral lesion detection, with 80% power and an alpha of 2.5% (one-sided). R version 4.2.0 (22 April 2022) was used for all statistical analyses[25].

## 3. Results

### 3.1. Biodistribution of [^68^Ga]Ga-DOTA-TATE Compared to Al[^18^F]F-NOTA-Octreotide

In this prospective study, 20 patients with biopsy-proven NET were enrolled to compare the biodistribution and clinical utility of [^68^Ga]Ga-DOTA-TATE and Al[^18^F]F-NOTA-Octreotide. No adverse events, adverse drug reactions or significant changes in vital signs were observed during the study. Both tracers showed similar physiological uptake in spleen, vascular pool and bone. However, [^68^Ga] Ga-DOTA-TATE exhibited significantly higher uptake in liver (*p* < 0.01), hypophysis (*p* < 0.01), salivary glands (*p* < 0.01), uncinate process (*p* < 0.05), adrenal glands (*p* < 0.01), kidneys (*p* < 0.05) and small intestine (*p* < 0.05). The highest uptake of Al[^18^F]F-NOTA-Octreotide was observed in the spleen, adrenal glands and kidneys, whereas a low uptake was observed for vascular pool, salivary glands and bone, a pattern similar to that seen with [^68^Ga]Ga-DOTA-TATE (Figure 1).

Tumour uptake of [^68^Ga]Ga-DOTA-TATE and Al[^18^F]F-NOTA-Octreotide was compared by means of SUV_max_ values and tumour-to-background ratios. [^68^Ga]Ga-DOTA-TATE showed higher uptake in liver (SUVmax: 8.76 ± 2.84 vs. 6.11 ± 2.24), hypophysis (SUVmax: 8.12 ± 3.05 vs. 5.75 ± 1.68), salivary glands (SUVmax: 4.1 ± 1.83 vs. 1.68 ± 0.29), uncinate process (SUVmax: 8.58 ± 2.91 vs. 6.82 ± 2.73), adrenal glands (SUVmax: 16.42 ± 4.8 vs. 12.75 ± 4.65), kidneys (SUVmax: 17.64 ± 4.01 vs. 13.41 ± 3.63) and small intestine (SUVmax: 5.73 ± 1.78 vs. 3.9 ± 1.33) as compared to Al[^18^F]F-NOTA-Octreotide. (Figure 1, Table 2).

[^68^Ga]Ga-DOTA-TATE showed higher tumour-to-liver ratios in lymph nodes metastasis (TLR: 3.8 ± 3.9 vs. 3.3 ± 2.3) and distant metastasis in lung, ovary, soft tissue and peritoneal carcinomatosis (TLR: 3.6 ± 6.0 vs. 3.1 ± 3.7) as compared to Al[^18^F]F-NOTA-Octreotide. On the contrary, metastatic lesions showed higher tumour-to-background ratios with Al[^18^F]F-NOTA-Octreotide in bone(TLR: 3.0 ± 1.8 vs. 2.1 ± 0.8) and primary tumour (TLR: 6.0 ± 2.9 vs. 4.8 ± 2.4, respectively) than with [^68^Ga]Ga-DOTA-TATE. We obtained similar results when calculating tumour-to-background ratios with spleen as reference organ. However, differences in TLR and TSR values were not statistically significant (Figure 2).

### 3.2. Tumoral Lesion Detection of [^68^Ga]Ga-DOTA-TATE Compared to Al[^18^F]F-NOTA-Octreotide

Next, we compared the number of lesions detected by [^68^Ga]Ga-DOTA-TATE and Al[^18^F]F-NOTA-Octreotide. Only four patients showed malignant lesions in the primary tumour site. For these patients, both tracers revealed the exact number of malignant lesions. While both tracers revealed liver metastasis in the same patients (17/20 patients), [^68^Ga]Ga-DOTA-TATE detected one additional metastatic lesion in patients No. 2, No. 14 and No. 15 and two additional lesions in patient No. 19 as compared to Al[^18^F]F-NOTA-Octreotide (Appendix A). In contrast, patient No. 10, showed one additional metastatic lesion with Al[^18^F]F-NOTA-Octreotide compared to [^68^Ga]Ga-DOTA-TATE (Figure 3). Despite these differences, all patients exhibited numerous metastatic lesions (Table 3) with a positive [^68^Ga]Ga-DOTA-TATE PET scan in total 366 metastatic lesions in the liver and Al[^18^F]F-NOTA-Octreotide PET detecting 362 lesions.

Hence, to determine a non-inferior detection of neoplastic lesions of Al[^18^F]F-NOTA-Octreotide PET compared to [^68^Ga]Ga-DOTA-TATE PET we employed a multilevel model since the results indicated evidence of clustering, confirmed by the correlation coefficient (0.68) and with a significant ANOVA test (*p* < 0.05). The results showed a mean difference, test-reference, of 0.074% (95% confidence interval: −3.874–4.022%) with a lower margin higher than the pre-specified boundary for non-inferiority (-5%), indicating that Al[^18^F]F-NOTA-Octreotide PET is non-inferior to [^68^Ga]Ga-DOTA-TATE PET (Figure 4).

## 4. Discussion

^68^Ga-labelled tracers for SSTR are the gold standard for imaging NET patients. However, countries with vast territorial areas and limited ^68^Ge/^68^Ga generators face an enormous logistical challenge. Al[^18^F]F-NOTA-Octreotide has emerged as an exciting alternative to ^68^Ga-tracers, especially due to the longer half-life of fluorine-18 compared to gallium-68 and production yield, facilitating distribution to distant clinical facilities. Moreover, fluorine-18 presents a shorter positron range resulting in an improved spatial resolution compared to gallium-68 [17].

To evaluate Al[^18^F]F-NOTA-Octreotide clinical utility, we performed a prospective study on 20 patients to compare [^68^Ga]Ga-DOTA-TATE versus Al[^18^F]F-NOTA-Octreotide. The biodistribution profile of Al[^18^F]F-NOTA-Octreotide and [^68^Ga]Ga-DOTA-TATE was comparable for the spleen showing high uptake with SUV_max_ values of 25.0 and 27.2, respectively. However, Al[^18^F]F-NOTA-Octreotide showed significantly less accumulation in the liver, hypophysis, salivary glands, uncinate process, adrenal gland, kidney, and small intestine compared to [^68^Ga]Ga-DOTA-TATE. The differences in uptake were most pronounced in salivary glands which is in line with previous studies reporting four to sixfold higher uptake [21]. Collectively our results revealed a high background uptake for [^68^Ga]Ga-DOTA-TATE compared to Al[^18^F]F-NOTA-Octreotide, which is consistent with previously published data [14]. However, while writing the present article, a new multicentric prospective study including a cohort of 75 NET patients histologically confirmed was published showing no significant differences in mean SUV_max_ in most organs [22]. Contrary to this report, we observed a higher mean SUV_max_ for [^68^Ga]Ga-DOTA-TATE compared to Al[^18^F]F-NOTA-Octreotide. This discrepancy may be due to the smaller group sample included in our study (20 patients). When the mean of the tumour-to-background ratio was analysed (using the liver, TLR, and spleen, TSR, as background tissue), non-significant differences were observed (Fig 2), demonstrating a lower liver and spleen background with Al[^18^F]F-NOTA-Octreotide compared to [^68^Ga]Ga-DOTA-TATE.

Previous studies have shown that both [^68^Ga]Ga-DOTA-TATE and Al[^18^F]F-NOTA-Octreotide are highly sensitive to detecting NET lesions. In fact, Hou et al., (2021) showed in a group of 20 patients that Al[^18^F]F-NOTA-Octreotide detected 177 lesions compared to 152 lesions with [^68^Ga]Ga-DOTA-TATE. This difference was particularly observed in the liver (116 vs. 93). Likewise, Pauwels, et al., 2022 showed that the detection ratio means of Al[^18^F]F-NOTA-Octreotide was significantly higher compared to [^68^Ga]Ga-DOTA-TATE/NOC (91.1% vs. 75.3% lesions). The differential detection ratio (calculated by the difference in detection ratio between Al[^18^F]F-NOTA-Octreotide and [^68^Ga]Ga-DOTA-TATE/NOC per each patient) was used to evaluate whether Al[^18^F]F-NOTA-Octreotide was non-inferior to [^68^Ga]Ga-DOTA-TATE and [^68^Ga]Ga-DOTA-NOC. This study concluded that Al[^18^F]F-NOTA-Octreotide was non-inferior compared to [^68^Ga]Ga-DOTA-TATE/NOC PET in NET patients [22].

Our study included patients with NETs (mainly G1/G2), which, in some cases, obstructed the lesions counting process. Thus, patients No. 7, No. 11, 13, 18, and 20 presented countless liver metastasis, which was registered as ≥50 lesions. This was also true for bone metastasis in the case of patients 11 and 13 (Table 2). In total, 748 lesions were detected (considering countless metastasis as at least 50 lesions), 751 with [^68^Ga]Ga-DOTA-TATE and 747 with Al[^18^F]F-NOTA-Octreotide.

In the liver, patients No. 2 [^68^Ga]Ga-DOTA-TATE detected 10 lesions compared to 9 detected by Al[^18^F]F-NOTA-Octreotide. Likewise, in the patient N°14 [^68^Ga]Ga-DOTA-TATE detected 20 lesions compared to 19 detected by Al[^18^F]F-NOTA-Octreotide, and in the patient No. 15 [^68^Ga]Ga-DOTA-TATE detected 16 lesions and Al[^18^F]F-NOTA-Octreotide only 15. In the case of patient No. 19, [^68^Ga]Ga-DOTA-TATE detected two more lesions compared to Al[^18^F]F-NOTA-Octreotide (13 vs. 11, respectively). In the case of patient No. 10, Al[^18^F]F-NOTA-Octreotide detected one more lesion than [^68^Ga]Ga-DOTA-TATE (3 vs. 2, respectively) (Figure 3). Interestingly, this patient presented a G3 NET tumour. This particular lesion was small, suggesting that ^18^F presents a superior capacity than [^68^Ga]Ga-DOTA-TATE to detect small lesions. This hypothesis is supported by the better spatial resolution of fluorine-18 compared to gallium-68 and by previous results with comparison using both tracers to detect < 5mm peritoneal metastasis [14,26].

The differences between both tracers in detecting liver lesions neither change the therapeutic approach nor the disease prognosis. In fact, none of these differences was statistically significant. In other metastatic lesions such as primary tumour, bone, lymph nodes, soft tissue, ovary lung or peritoneal carcinomatosis, [^68^Ga]Ga-DOTA-TATE and Al[^18^F]F-NOTA-Octreotide had comparable effectiveness, detecting the same number of lesions.

Our results confirm that Al[^18^F]F-NOTA-Octreotide is not inferior to [^68^Ga]Ga-DOTA-TATE PET to detect lesions in NET patients since the lower margin of the 95% of the confidence interval was higher than the lower pre-specified boundary of −5% for non-inferiority (Figure 4). Thus, our results confirmed previous results [14,22] in Latin-American NET patients in which the distribution and production of radiotracers is a crucial challenge, especially considering the geography and prevalence of cancer [27].

Our study found non-significant differences in detection rates between Al[^18^F]F-NOTA-Octreotide and [^68^Ga]Ga-DOTA-TATE PET. However, one limitation of our study is the small subgroup of patients included and missing complementary data such as other imaging modalities, such as magnetic resonance imaging or ^18^F-FDG PET/CT imaging. Ongoing research at our centre is now focused on evaluating the relationship between the tumoral grade and lesion uptake in larger groups of patients.

## 5. Conclusions

Our study is the first, to our knowledge, to be performed on Latin-American NET patients. Our results show that Al[^18^F]F-NOTA-Octreotide exhibits a similar biodistribution to that of [^68^Ga]Ga-DOTA-TATE, with similar detection rates in different organs, demonstrating the non-inferiority of Al[^18^F]F-NOTA-Octreotide-PET/CT compared to [^68^Ga]Ga-DOTA-TATE-PET/CT. Therefore, Al[^18^F]F-NOTA-Octreotide is an important alternative for NET patients, especially in countries (such as Chile) with vast territories and limited ^68^Ge/^68^Ga generators. Future studies should include more patients to evaluate the clinical utility of Al[^18^F]F-NOTA-Octreotide for staging NET cancer patients and evaluate the potential detection of small and less differentiated lesions.

## Figures and Tables

**Figure 1 cancers-15-00439-f001:**
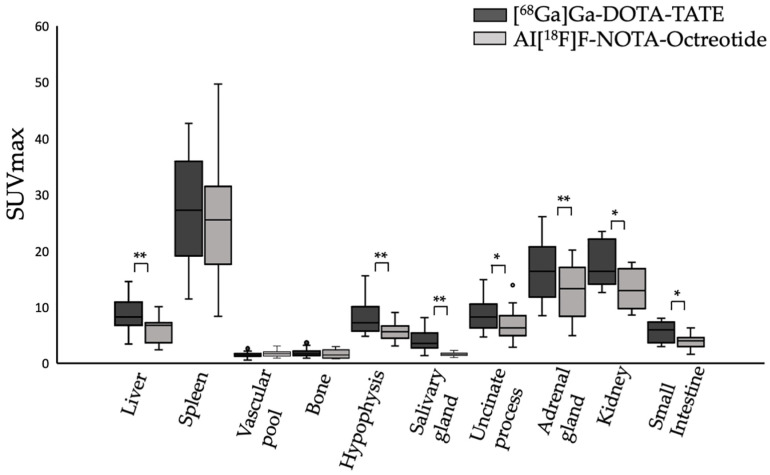
Physiological uptake (SUVmax) of [^68^Ga]Ga-DOTA-TATE (black bars) and Al[^18^F]F-NOTA-Octreotide (grey bars) in different tissues. * (*p* < 0.05), ** (*p* < 0.001).

**Figure 2 cancers-15-00439-f002:**
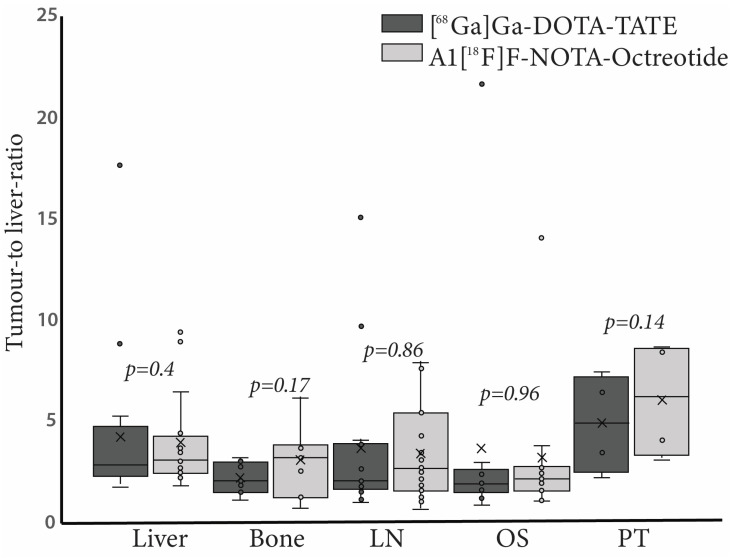
Tumour-to-liver ratio (TLR) of [^68^Ga]Ga-DOTA-TATE (black bars) and Al[^18^F]F-NOTA-Octreotide (grey bars) in different tissues. Right panel, boxplots representing TLR in liver, bone, lymph node (LN), other sites (OS) and primary tumour (PT). In each case outliers are represented as circles.

**Figure 3 cancers-15-00439-f003:**
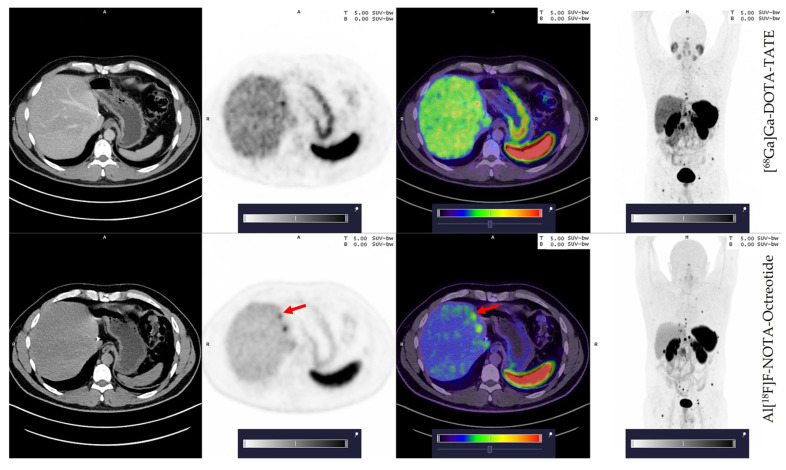
Patient 10; 42 years old, male with an additional small liver lesion (red arrow) detected by Al[^18^F]F-NOTA-Octreotide (lower row) and not seen with [^68^Ga]Ga-DOTA-TATE (upper row). Colour scale bar representing SUV values ranging from 0.0–5.0.

**Figure 4 cancers-15-00439-f004:**
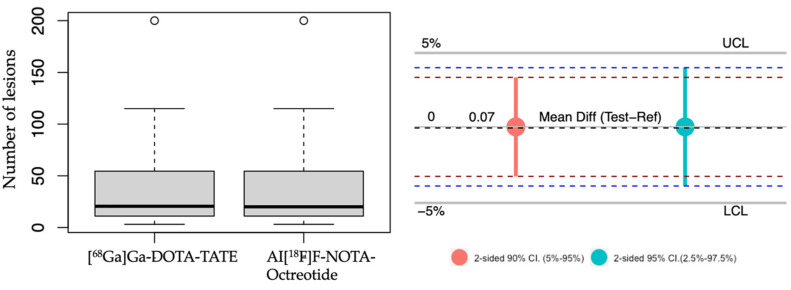
Non-inferiority analysis. Left panel, boxplot representing the average of tumoral lesions detected with [^68^Ga]Ga-DOTA-TATE and Al[^18^F]F-NOTA-Octreotide. right panel, a non-inferiority chart 2-sided 90% confidence interval (red) and 95% confidence interval (light blue). UCL: upper clinically significant margin limit 5%. LCL: lower clinically significant margin limit of 5%. The mean difference is indicated. Outliers are represented as circles.

**Table 1 cancers-15-00439-t001:** Patients’ characteristics.

Patient ID	Age (y)	Gender	Primary Tumour	Tumour Grade	Ki-67 Index	[^68^Ga]Ga-DOTA-TATE Activity (MBq)	Al[^18^F]F-NOTA-Octreotide Activity (MBq)	Delay (Days)
1	73	M	Small Intestine	G2	6%	156.1	262.3	7
2	45	F	Bronchial	G1	1%	152.1	185.0	7
3	49	M	Pancreas	G2	8%	126.2	272.7	2
4	44	M	Pancreas	G2	5%	149.5	282.7	2
5	73	M	Apendice cecal	G1	1%	141.3	301.9	23
6	69	M	Small Intestine	G2	5%	165.8	265.3	23
7	60	M	Small Intestine	G1	1%	159.8	276.0	30
8	58	M	Small Intestine	G1	1%	95.8	284.9	23
9	40	M	Small Intestine	G2	12%	179.8	214.6	6
10	42	M	Liver	G3	>20%	143.2	236.8	22
11	57	F	Unknown	G3	30%	146.2	251.6	8
12	57	F	Colon	G2	15%	172.4	254.2	7
13	60	F	Small Intestine	G1	2%	172.1	232.3	7
14	54	F	Small Intestine	G2	NA	97.7	256.8	13
15	63	F	Small Intestine	G2	5%	165.8	163.9	8
16	60	M	Unknown	G2	NA	131.0	237.1	8
17	52	M	Small Intestine	G1	1%	175.0	254.9	13
18	55	M	Small Intestine	G2	5%	145.0	245.6	15
19	83	F	Small Intestine	G2	3%	152.8	252.0	16
20	52	M	Small Intestine	G2	3%	153.5	205.7	15

**Table 2 cancers-15-00439-t002:** [^68^Ga]Ga-DOTA-TATE PET and Al[^18^F]F-NOTA-Octreotide PET SUV_max_ values.

Organ	[^68^Ga]Ga-DOTA-TATE SUV_max_	Al[^18^F]F-NOTA-Octreotide SUV_max_	*p*-Value
Liver	8.76 ± 2.83	6.11 ± 2.23	<0.001
Spleen	27.18 ± 9.91	25 ± 10.91	0.247
Vascular pool	1.543 ± 0.62	1.71 ± 0.54	0.07
Bone	1.86 ± 0.68	1.66 ± 0.74	0.262
Hypophysis	8.11 ± 3.05	5.75 ± 1.68	<0.001
Salivary gland	4.1 ± 1.83	1.61 ± 0.29	<0.001
Uncinate process	8.58 ± 2.91	6.82 ± 2.73	<0.05
Adrenal gland	16.42 ± 4.8	12.75 ± 4.65	<0.001
Kidney	17.64 ± 4	13.4 ± 3.63	<0.05
Small intestine	5.73 ± 1.88	3.9 ± 1.33	<0.05

**Table 3 cancers-15-00439-t003:** Neoplastic lesions detected with [^68^Ga]Ga-DOTA-TATE and Al[^18^F]F-NOTA-Octreotide.

Patients	Primary Tumour		Liver Metastases **		Bone Metastases		LN Metastases		Other Sites Metastases		Total Lesions	
Tracer *	GAD	FOC	GAD	FOC	GAD	FOC	GAD	FOC	GAD	FOC	GAD	FOC
1	0	0	1	1	15	15	10	10	0	0	26	26
2	1	1	**10**	**9**	6	6	1	1	0	0	18	17
3	1	1	10	10	0	0	0	0	0	0	11	11
4	1	1	5	5	0	0	0	0	0	0	6	6
5	0	0	27	27	3	3	25	25	1	1	≥50	≥50
6	1	1	0	0	0	0	11	11	1	1	13	13
7	0	0	≥50	≥50	0	0	3	3	0	0	≥50	≥50
8	0	0	2	2	0	0	4	4	1	1	7	7
9	0	0	0	0	0	0	4	4	0	0	4	4
10	0	0	**2**	**3**	16	16	2	2	3	3	23	24
11	0	0	≥50	≥50	≥50	≥50	0	0	0	0	≥50	≥50
12	0	0	8	8	0	0	0	0	3	3	11	11
13	0	0	>50	>50	≥50	≥50	13	13	2	2	≥50	≥50
14	0	0	**20**	**19**	0	0	1	1	3	3	24	23
15	0	0	**16**	**15**	0	0	1	1	1	1	18	17
16	0	0	0	0	11	11	4	4	2	2	17	17
17	0	0	2	2	0	0	1	1	0	0	3	3
18	0	0	≥50	≥50	0	0	31	31	≥50	≥50	≥50	≥50
19	0	0	**13**	**11**	1	1	1	1	≥50	≥50	≥50	≥50
20	0	0	≥50	≥50	0	0	0	0	0	0	≥50	≥50

* GAD: [^68^Ga]Ga-DOTA-TATE; FOC: Al[^18^F]F-NOTA-Octreotide. ** Patients with discordant lesions are highlighted in bold. Patients where [^68^Ga]Ga-DOTA-TATE detected more lesions are highlighted in blue, where Al[^18^F]F-NOTA-Octreotide detected more lesions are highlighted in yellow.

## Data Availability

The data presented in this study are available on request from the corresponding author. The data are not publicly available due to restrictions regarding privacy of patients and ethical reasons.

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
