# Peer review of "Al[18F]F-NOTA-Octreotide Is Comparable to [68Ga]Ga-DOTA-TATE for PET/CT Imaging of Neuroendocrine Tumours in the Latin-American Population"

_cancers, 2023, doi:10.3390/cancers15020439_

Round 1

Reviewer 1 Report

This manuscript compares Al[18F]F-NOTA-Octreotide and [68Ga]Ga-DOTA-TATE for PET/CT imaging of neuroendocrine tumors, specifically in the Latin-American population. It is a meaningful topic to promote the SSTR-PET application in developing countries. The results are compared with similar clinical data reported elsewhere, and the major conclusions are almost identical. Other comments are as below:

  1. Since most patients are at G1 and G2 grades, “advanced” is recommended to be removed from the title. 
  2. Page 2, line 62: Change Fluor-18 to Fluorine-18. 
  3. Page 2, the paragraph from line 69: It is recommended to better summarize the literature data for reported comparison between the two PET tracers.
  4. Method 2.2: What was the dosing protocol? Patients #8 and #14 received low 68Ga doses but relatively high 18F doses. 
  5. Figure 2 needs to be in better resolution. 
  6. Page 5, line 146: Since this paragraph describes the results based on the quantitative SUV data comparison, it is recommended to use tumor-to-non-tumor ratios instead of “imaging contrast”. TLR has been defined as tumor-to-liver ratio, but herein it is also used to describe tumor-to-lymph nodes ratio. Please correct it. 
  7. Page 8: Please spell out MRI at its first-time showing in the manuscript. Please double-check if there are any other needs in spelling out abbreviations. 
  8. Page 8, line 265: Please correct exhibit with exhibits. Please check for other minor grammar errors.    

Author Response

Dear Reviewer 1, thanks for your comments. Please see our point-by-point response in the attachment.

Reviewer 2 Report

The manuscript “Al[18F]F-NOTA-Octreotide is comparable to [68Ga]Ga-DOTA-TATE for PET/CT imaging of advanced neuroendocrine tumors in the Latin-American population” re[ports an interesting study on PET imaging of neuroendocrine tumors (NET) using Al[18F]F-NOTA-Octreotide and comparing with [68Ga]Ga DOTA-TATE. The study demonstrates that Al[18F]F-NOTA-Octreotide provided excellent image quality, visualized NET lesions with high sensitivity and represents a highly promising, clinical alternative to standard clinical agent [68Ga]Ga-DOTA-TATE. The results are interesting for future use of Al[18F]F-NOTA-Octreotide. 

1.     Authors should provide molecular structure of both [68Ga]Ga DOTA-TATE and Al[18F]F-NOTA-Octreotide in Figure 1 for readers’ understanding. These molecular structures are available online.

2.     Figure 3; scale bar should be labelled. [68Ga]Ga-DOTA-TATE images show a lesion distinguished as red spot and another lesion is not distinguishable, however, Al[18F]F-NOTA-Octreotide shows two lesion with both green spots. The liver is green in case of [68Ga]Ga-DOTA-TATE due to higher uptake? And the liver is blue in case of Al[18F]F-NOTA-Octreotide due to lower uptake? (if both scales are same). Please clarify.

“Both radiotracers showed high spleen and liver uptake (mean SUVmax)” line 199.

Author Response

Dear Reviewer 2, thanks for your comments. Please see our point-by-point response in the attachment.

Round 2

Reviewer 2 Report

This revised version of the manuscript responses reviewers' comments satisfactorily. I have no further comments, and recommend for publication.